# A pan-orthohantavirus human lung xenograft mouse model and its utility for preclinical studies

Melanie Rissmann[1,*,☾], Danny Noack[1,☾], Thomas M. Spliethof[2], Vincent P. Vaes[1], Rianne Stam[1], Peter van Run[1], Jordan J. Clark[3,4], Georges M. G. M. Verjans[1], Bart L. Haagmans[1], Florian Krammer[3,4,5], Marion P. G. Koopmans[1], Judith M. A. van den Brand[2], Barry Rockx[1,*]

1 Department of Viroscience, Erasmus University Medical Center, Rotterdam, the Netherlands, 2 Division of Pathology, Faculty of Veterinary Medicine, Utrecht University, Utrecht, the Netherlands, 3 Department of Microbiology, Icahn School of Medicine at Mount Sinai, New York, New York, United States of America, 4 Center for Vaccine Research and Pandemic Preparedness (C-VaRPP), Icahn School of Medicine at Mount Sinai, New York, New York, United States of America, 5 Department of Pathology, Icahn School of Medicine at Mount Sinai, New York, New York, United States of America

☾ These authors contributed equally to this work.
*m.rissmann@erasmusmc.nl (MR); b.rockx@erasmusmc.nl (BR)

## Abstract

Orthohantaviruses are emerging zoonotic viruses that can infect humans via the respiratory tract. There is an unmet need for an *in vivo* model to study infection of different orthohantaviruses in physiologically relevant tissue and to assess the efficacy of novel pan-orthohantavirus countermeasures. Here, we describe the use of a human lung xenograft mouse model to study the permissiveness for different orthohantavirus species and to assess its utility for preclinical testing of therapeutics. Following infection of xenografted human lung tissues, distinct orthohantavirus species differentially replicated in the human lung and subsequently spread systemically. The different orthohantaviruses primarily targeted the endothelium, respiratory epithelium and macrophages in the human lung. A proof-of-concept preclinical study showed treatment of these mice with a virus neutralizing antibody could block Andes orthohantavirus infection and dissemination. This pan-orthohantavirus model will facilitate progress in the fundamental understanding of pathogenesis and virus-host interactions for orthohantaviruses. Furthermore, it is an invaluable tool for preclinical evaluation of novel candidate pan-orthohantavirus intervention strategies.

## Author summary

Orthohantaviruses are rodent-borne pathogens that can be transmitted from rodents to humans by inhalation of aerosolized virus-contaminated rodent excreta. While orthohantaviruses generally do not cause overt clinical signs in rodents, orthohantavirus infection can lead to severe disease with case fatality rates up to 40%. The pathogenesis of associated hemorrhagic fever with renal syndrome (HFRS) mainly in Europe and Asia and hantavirus cardiopulmonary syndrome (HCPS) in the Americas remains poorly

**Data availability statement:** The data that support this study are available in the article and its supplementary files.

**Funding:** This work was funded in part by the Netherlands Centre for One Health Ph.D. Research Program (http://www.ncoh.nl) to DN. This work was furthermore funded by the Health~Holland grant (LSHM19136), co-funded by the PPP Allowance made available by the Health~Holland, Top Sector Life Sciences & Health, to stimulate public–private partnerships. The funders had no role in study design, data collection and analysis, decision to publish, or preparation of the manuscript.

**Competing interests:** I have read the journal's policy and the authors of this manuscript have the following competing interests: The Icahn School of Medicine at Mount Sinai has filed patent applications relating to SARS-CoV-2 serological assays, NDV-based SARS-CoV-2 vaccines, influenza virus vaccines and influenza virus therapeutics which list FK as co-inventor and of which several have been licensed. Mount Sinai has spun out a company, Kantaro, to market serological tests for SARS-CoV-2 and another company, Castlevax, to develop SARS-CoV-2 vaccines. FK is co-founder and scientific advisory board member of Castlevax. FK has consulted for Merck, Curevac, Seqirus and Pfizer and is currently consulting for 3rd Rock Ventures, GSK, Gritstone and Avimex. The Krammer laboratory is also collaborating with Dynavax on influenza vaccine development. All other authors have no conflicts of interest to report.

understood. Particularly the understanding of the early steps of pathogenesis occurring in the human lungs is limited. In addition, assessment of the efficacy of cross-reactive anti-orthohantavirus antibodies is severely hampered by the limited availability of *in vivo* models that allow for replication of both HFRS- and HCPS-associated orthohantaviruses within the same animal model. Therefore, we present a human lung xenograft mouse model to study early events of orthohantavirus pathogenesis in the human lung. We demonstrated that multiple orthohantaviruses replicate inside the human lung xenografts, and characterized the tropism and host responses within these human lung xenografts. Also, as a proof-of-principle, we showed that the model can be utilized to assess the protective efficacy of an existing monoclonal antibody against Andes virus infection. Overall, this pan-orthohantavirus model allows for the study of orthohantavirus pathogenesis and will aid as a novel platform to test anti-orthohantavirus countermeasures.

## Introduction

Orthohantaviruses are zoonotic viruses, which are naturally carried by rodents and can be transmitted to humans. Transmission primarily occurs via inhalation of aerosolized viruses present in rodent excreta, such as saliva, urine and feces. However, human-to-human transmission has been described for one of the orthohantavirus species, Andes orthohantavirus (ANDV) [1]. Upon initial infection in the human respiratory tract, these viruses can subsequently cause severe symptoms such as hantavirus cardiopulmonary syndrome (HCPS) and hemorrhagic fever with renal syndrome (HFRS). Although it is known from lethal HCPS infections in humans that pulmonary endothelial cells, epithelial cells and macrophages are infected during disease [2,3], there is a lack of understanding which cells of the respiratory tract are initially infected and which early events lead to systemic dissemination of orthohantaviruses to endothelial cells in different organs throughout the human body [4]. Knowledge on the early mechanisms of pathogenesis in the human lung is also currently limited due to scarce availability of acute clinical samples and relevant animal models that recapitulate infection in the human lung.

Although orthohantaviruses enter the human body upon inhalation of virus particles, the role of respiratory epithelium during early stages of infection remains largely unexplored. The specific sites of initial infection and the molecular mechanisms that contribute to disease within the human respiratory tract remain unidentified. Current *in vitro* models are limited as they do not reflect the complexity of the human lung, which contains up to forty different cell types including but not limited to respiratory epithelial, endothelial, mesenchymal and smooth muscle cells [5]. Furthermore, a local microenvironment and circulation may be essential factors of the initial stages of pathogenesis of these predominantly endotheliotropic viruses.

Although pathogenesis is poorly understood, a strong neutralizing antibody response is suggested to be an important factor for protection [6,7]. Recent advances have therefore been made to isolate neutralizing monoclonal antibodies targeting orthohantaviruses [8–11] and to characterize pan-orthohantavirus neutralizing human antibodies as potential therapeutics [12,13]. However, the ability to confirm the breadth of the neutralizing capacities of such antibodies is hampered by the lack of a pan-orthohantavirus preclinical animal model that allows replication and study of multiple orthohantaviruses in the same model. As a consequence, there are currently no HCPS and/or HFRS vaccines or therapeutics approved by the United States Food and Drug Administration (FDA) or European Medicines Agency (EMA).

Previous studies have attempted to develop relevant animal models to study human orthohantavirus infection. These studies were conducted in non-human primates (NHPs), (genetically-modified) laboratory mice, rats, hamsters and ferrets [14–20]. However, only NHPs and (immunosuppressed) Syrian hamsters allow for disseminated infection of both HCPS–[20–27] and HFRS-[28–33] associated viruses. Inoculation of HFRS-associated viruses in various tested animal models, such as ferrets, results in an animal infection model in absence of clinical signs or even detectable viremia [18]. While NHP models are restricted by ethical and economic concerns, the immunocompetent Syrian hamster model only displays typical symptoms observed in humans following ANDV inoculation [27] and not following inoculation of other highly pathogenic orthohantaviruses, such as Sin Nombre orthohantavirus (SNV) [26] or Hantaan orthohantavirus (HTNV) [34]. The hamster model is also restricted by limited availability of species-specific immunological reagents, but more importantly, must be interpreted carefully due to limited translational potential for the study of relevant human tissues. Yet none of the known animal models were utilized to study broadly reactive pan-orthohantavirus intervention strategies due to limitations of the breadth of permissiveness for different orthohantavirus species.

Previously, we developed a human lung xenograft mouse model based on severely immunocompromised NOD.Cg-Prkdc$^{scid}$Il2rg$^{tm1Wjl}$/SzJ (NSG) mice to study Nipah virus infection and early host responses in highly relevant tissue [35]. This approach was subsequently shown to be useful for other viruses as well, including severe acute respiratory syndrome coronavirus 2 (SARS–CoV-2) [36], Middle East respiratory syndrome coronavirus (MERS-CoV), respiratory syncytial virus (RSV) and human cytomegalovirus (HCMV) [37].

Here, we describe a pan-orthohantavirus mouse model that allows for study of viral entry and infection with different orthohantaviruses in relevant human lung tissue. The possibility to utilize the same animal model for infection with different HCPS- and HFRS-associated orthohantaviruses ultimately allows for preclinical investigations of future cross-protective therapeutics against different orthohantaviruses. As a proof-of-principle of this novel preclinical platform, we demonstrated the protective effects of a previously described neutralizing monoclonal antibody against ANDV [8]. This model represents a valuable contribution to the understanding of early events in orthohantavirus infection in the human lungs, as well as for the development of novel pan-orthohantavirus therapeutics.

## Results

### Establishment of a human lung xenograft mouse model

Human fetal lung tissue was implanted into dorsal subcutaneous pockets of NSG mice with a subsequent maturation period of minimal 12 weeks (Fig 1a), during which easily accessible human lung xenografts developed with a success rate of more than 80%, i.e., 80% of implanted tissues evolving into mature xenografts. Macroscopically, human lung xenografts grew to visible and palpable masses of up to 1 cm circumference. Xenografts were highly vascularized and encapsulated (Fig 1b). Microscopically, human lung xenografts were composed of structures similar to those seen in adult human lung, and included cartilage, blood vessels, ciliated pseudostratified columnar epithelium, and primitive "air" spaces filled with mucus and lined by cuboidal to flat epithelium (Fig 1c). The specific presence of endothelial and respiratory epithelial cells as well as resident immune cells as main target cells for orthohantaviruses, was confirmed by immunostaining (Fig 1d). Additionally, the presence of three entry receptors and co-factors (protocadherin-1, β$_3$ integrin and CD55), which are currently suggested to be involved in orthohantavirus binding and entry [38–40], was confirmed (Fig 1e). In correspondence with earlier findings, protocadherin-1 was found to be abundantly expressed on

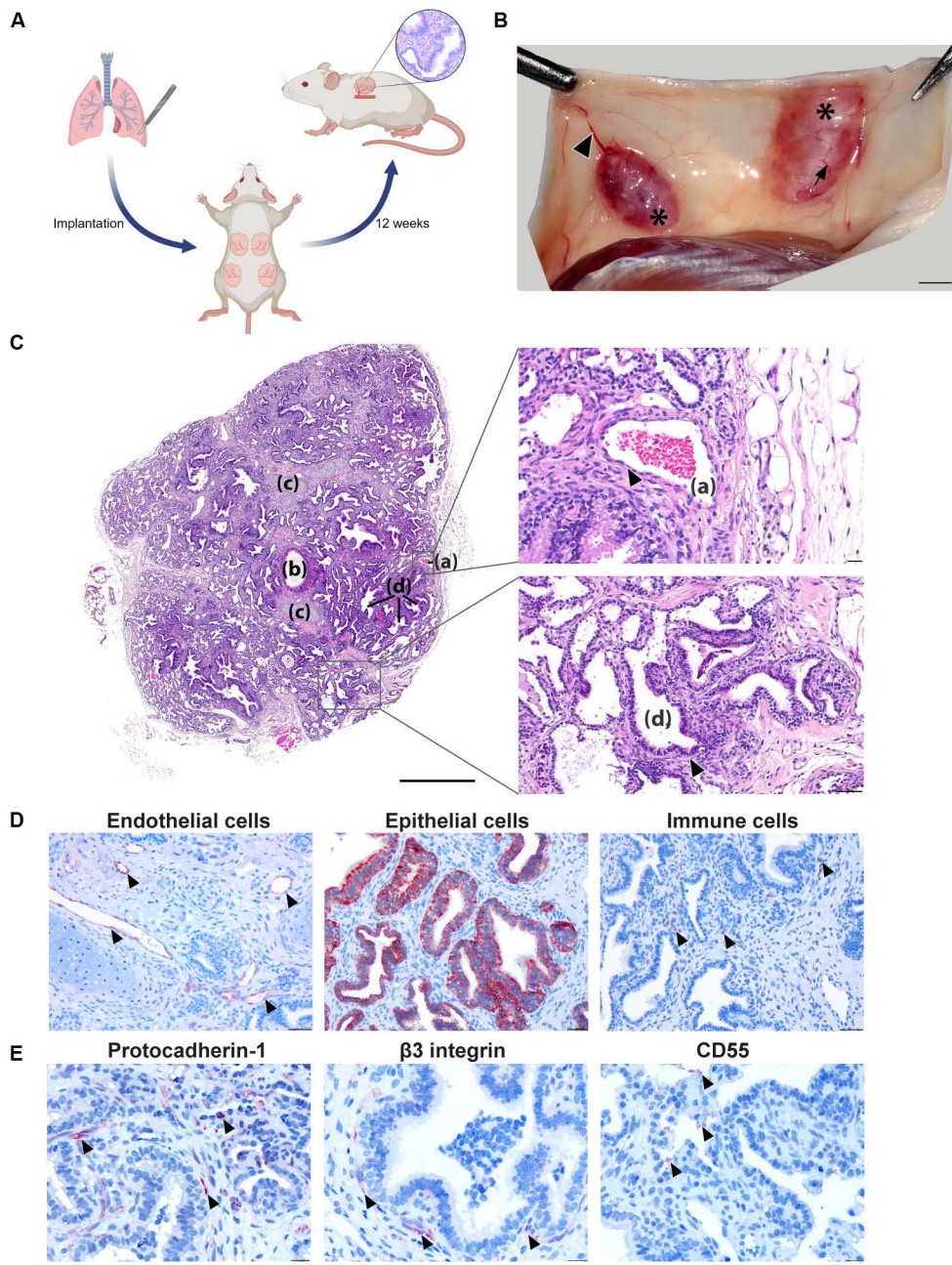

**Fig 1. Human lung xenograft development and characterization. a)** Human fetal lung tissues were implanted into four separate subcutaneous pockets on the back of at least 8 weeks old NOD.Cg-Prkdc$^{scid}$Il2rg$^{tm1Wjl}$/SzJ (NSG) mice. These xenografts were left for a maturation period of at least 12 weeks. *Created in BioRender. 1, V. (2025)* https://BioRender.com/r28y992. **b)** Macroscopic appearance of human lung xenografts, after longitudinal dissection and lifting of the murine skin. The human lung tissues developed into encapsulated and highly vascularized (arrow) human lung xenografts (*) that are connected to the blood circulation of the mouse (arrow head). Scale bar represents 1 cm. **c)** Histological appearance of human lung xenografts. The xenograft is composed of blood vessels (a), cartilaginous airways (b), cartilage (c) and non-cartilaginous airways (d). Scale bar represents 800 μm. Top zoom-in shows endothelial cells (indicated by arrow head) lining a blood vessel. Scale bar represents 20 μm. Bottom zoom-in shows respiratory, partially ciliated (indicated by arrow head), epithelium lining a non-cartilaginous airway. Scale bar represents 50 μm. **d)** Immunohistochemistry staining for endothelial cells (Von Willebrand factor+, indicated by arrow heads), respiratory epithelial cells (cytokeratin 19+) and immune cells (CD45+, indicated by arrow heads). Scale bars represent 20 μm. **e)** Immunohistochemistry staining for orthohantavirus entry (co-)receptors protocadherin-1, β3 integrin and CD55 as indicated by arrow heads in each image. Scale bar represents 20 μm.

endothelial and respiratory epithelial cells [39,41], $\beta_3$ integrin localized mainly on endothelial and epithelial cells [42], and CD55 was present on epithelial cell surfaces and adjacent extracellular fluids and immune cells [43]. Altogether, implantation led to successful engraftment and vascularization of human lung xenografts, composed of mature structures of the human respiratory tract and physiologically relevant cell types with presence of receptors involved in orthohantavirus infection.

## Both HCPS- and HFRS-associated orthohantaviruses replicate in the human lung xenograft mouse model

Human orthohantavirus infection occurs via inhalation of infectious particles, rendering the respiratory tract the first location of viral exposure. Therefore, the replicative potential of four distinct orthohantavirus species in human lung tissue was studied within this model (Fig 2). For each mouse, half of the human lung xenografts (maximum of two) were directly inoculated following intragraft inoculation. To examine if systemic viral dissemination could serve as an alternative infection route of orthohantaviruses in this model, the remaining xenografts were left untreated (non-inoculated). On 1, 3, 10 and 21 days post inoculation (dpi), animals (N = 6) from each experimental group were sacrificed. Two additional control groups were included in which animals were sacrificed at the end of the study at 21 dpi.

To study HCPS-associated viruses, ANDV and SNV were included. Both viruses efficiently replicated in directly inoculated xenografts following intragraft inoculation (Fig 3a). ANDV titers increased over time to peak titers of 5.36 Log$_{10}$ TCID$_{50}$/g (mean; range 0–6.30 Log$_{10}$ TCID$_{50}$/g) at 21 dpi, whereas SNV titers reached a peak of 3.98 Log$_{10}$ TCID$_{50}$/g (mean; range 0–6.58 Log$_{10}$ TCID$_{50}$/g) at 10 dpi after an initial reduction due to remaining inoculum at the first time point. To determine if this set-up could be utilized as a pan-orthohantavirus model, HTNV and Seoul orthohantavirus (SEOV) were included as prototypical HFRS-associated viruses. In the directly inoculated human lung xenografts, HTNV replicated with a plateau of

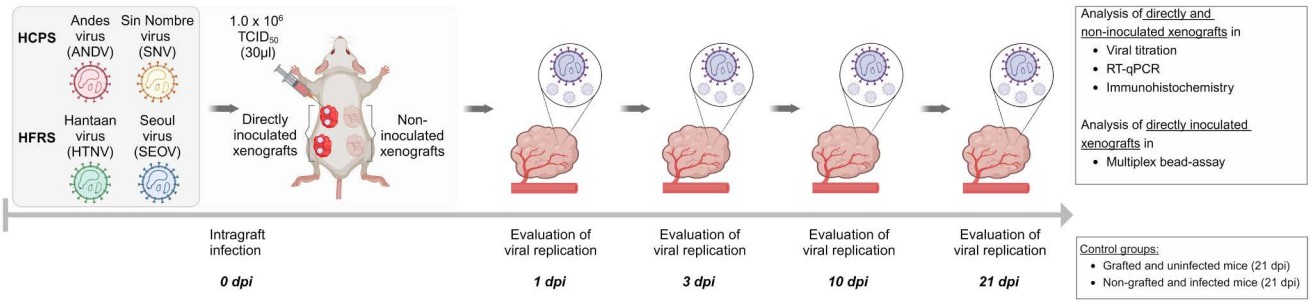

**Fig 2. Experimental set-up of the human lung xenograft mouse model.** The replicative potentials of Andes orthohantavirus (ANDV), Sin Nombre orthohantavirus (SNV), causative agents of hantavirus cardiopulmonary syndrome (HCPS), and Hantaan orthohantavirus (HTNV) and Seoul orthohantavirus (SEOV), causative agents of hemorrhagic fever with renal syndrome (HFRS), were evaluated in the human lung xenograft mouse model. In each of the 24 animals per virus, a maximum of two human lung xenografts were directly inoculated (intragraft inoculation) with 30 µl of 1.0 x 10$^6$ TCID$_{50}$ of ANDV, SNV, HTNV or SEOV. To examine if systemic viral dissemination could serve as an alternative infection route of orthohantaviruses, all remaining (maximum of two) xenografts were left non-inoculated. Following inoculation, the body weight of all animals was monitored regularly. On 1, 3, 10 and 21 days post inoculation (dpi), animals (N = 6) from each experimental group were euthanized. During all necropsies, the xenografts, murine lungs, liver, kidney, spleen and serum were collected for quantification of viral RNA load as well as histopathological examination. Additionally, directly inoculated xenografts of 10 and 21 dpi were analyzed by multiplex bead-assay. Two additional control groups were included in which animals were sacrificed at 21 dpi. Grafted, but non-inoculated animals served as control for human lung tissue composition. Non-grafted animals were subcutaneously inoculated with 30 µl of 1.0 x 10$^6$ TCID$_{50}$ as controls for susceptibility of the NSG mice to orthohantavirus infection. *Created in BioRender. 1, V. (2025)* https://BioRender.com/i75m985.

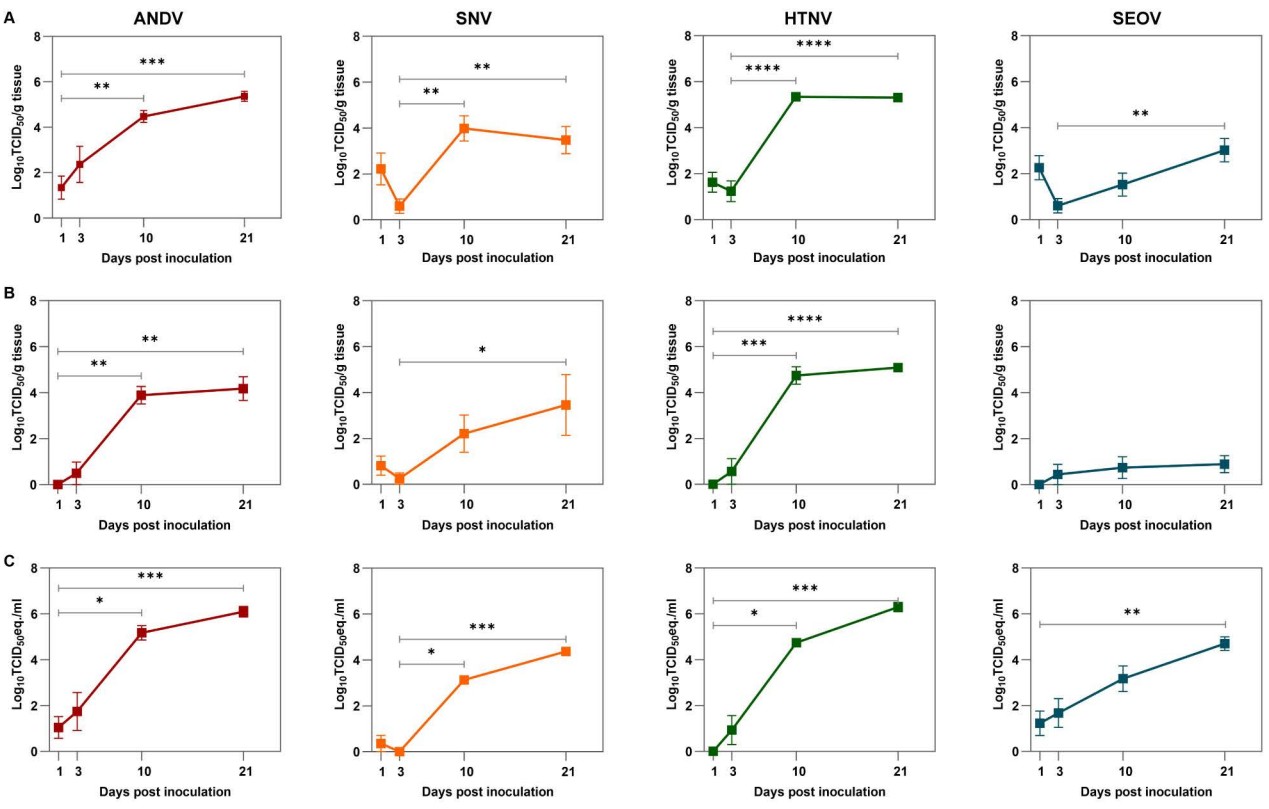

**Fig 3. Replication kinetics of distinct orthohantaviruses. a)** Infectious viral titers were quantified in human lung xenografts that were directly inoculated with ANDV, SNV, HTNV and SEOV. **b)** Infectious viral titers were determined in non-inoculated xenografts. Squares indicate the mean $TCID_{50}$ per gram tissue and error bars represent the standard error of the mean. **c)** Viral RNA was detected in serum via RT-qPCR. Squares indicate the mean $TCID_{50}$ equivalents per ml and error bars represent the standard error of the mean. Infectious titers in directly inoculated xenografts (a), non-inoculated xenografts (b) and viral RNA in serum (c) were compared to the lowest infectious titer (a, b) or viral RNA load (c) during the course of infection, i.e., 1 or 3 days post inoculation (dpi) by Kruskall-Wallis test with Dunn's multiple comparisons test. *p < 0.05, **p < 0.005, ***p < 0.001, ****p < 0.0001. Six animals were included per virus per time point.

5.34 $Log_{10}$ $TCID_{50}$/g (mean; range 3.87 $Log_{10}$–6.05 $Log_{10}$ $TCID_{50}$/g) at 10 dpi until the end of the experiment (Fig 3a). SEOV titers initially demonstrated a minor decrease in inoculated human lung xenografts due to remaining inoculum detected at 1 dpi, but then increased up to 3.02 $Log_{10}$ $TCID_{50}$/g (mean; range 0 $Log_{10}$–5.45 $Log_{10}$ $TCID_{50}$/g) at 21 dpi. In contrast to virus replication due to direct intragraft inoculation (Fig 3a), non-inoculated xenografts became infected following systemic dissemination of virus (Fig 3b). Hence, viremia was confirmed for all four viruses and followed similar temporal kinetics as viral replication in the non-inoculated xenograft tissues (Fig 3c). Thus, upon dissemination of ANDV from directly inoculated xenografts, viral titers reached a peak of 4.18 $Log_{10}$ $TCID_{50}$/g (mean; range 2.68 $Log_{10}$–4.96 $Log_{10}$ $TCID_{50}$/g) at 21 dpi, while SNV reached a peak of 3.46 $Log_{10}$ $TCID_{50}$/g (mean; range 0 –5.74 $Log_{10}$ $TCID_{50}$/g) at 21 dpi in the non-inoculated xenografts. The same trend was observed for HTNV with peak titers of 5.09 $Log_{10}$ $TCID_{50}$/g (mean; range 4.03 $Log_{10}$–5.86 $Log_{10}$ $TCID_{50}$/g) at 21 dpi and for SEOV peaking at 0.89 $Log_{10}$ $TCID_{50}$/g (mean; 0–2.83 $Log_{10}$ $TCID_{50}$/g) at 21 dpi. Furthermore, replication of all four orthohantaviruses was confirmed by presence of viral RNA in the directly inoculated (S1a Fig) and non-inoculated (S1b Fig) xenografts. Additionally, as an initial pilot experiment for the model, the maximum number

of available human lung xenografts per mouse were inoculated with Puumala orthohantavirus (PUUV), an orthohantavirus associated with milder clinical outcome. The broad applicability of our pan-orthohantavirus model was also confirmed by the increase of PUUV in human lung xenografts over time (S2a and S2b Fig). None of the animals in the entire study demonstrated any clinical symptoms or changes in body weight after infection (S3 Fig).

For comprehensive characterization of this pan-orthohantavirus model, viral spread throughout the murine organism was also monitored. In the murine lungs, which were highly susceptible due to the severely immunocompromised genetic background of the mice, all viruses reached peak titers at 21 dpi (S4a and S4b Fig). In the other tested murine organs, i.e., kidneys, liver and spleen, viral RNA loads were generally detected with higher levels at later time points (S4c Fig). In addition, inoculation of a control group of non-grafted mice showed that while these mice were susceptible to infection with orthohantaviruses, this infection resulted in less efficient systemic viral spread and increased variation throughout the murine organs (S5a and S5b Fig). Finally, we observed only minor variation in orthohantavirus replication kinetics based on the human fetal donor background of the lung tissue (S6a Fig) and sex of the mice (S6b Fig). Overall, this is a robust model that allowed for initial virus replication within the directly inoculated human lung xenografts with subsequent viremia and viral spread to other human lung xenografts and murine organs for all tested orthohantaviruses without causing clinical signs in these animals.

## Orthohantavirus cell tropism and histopathology in human lung xenografts

To assess the extent of infection and histopathological changes associated with orthohantavirus infections in the human lung, both directly inoculated and non-inoculated xenografts were evaluated in immunohistochemistry (IHC) for virus antigen and for histopathology in the hematoxylin and eosin (H&E) staining. Cells with a positive IHC signal for orthohantavirus antigen were detected for all viruses at all time points, however to variable extents for each experimental group at different time points (Figs 4a, 4b, 4c, S2c and S2d). ANDV antigen was detected in human lung tissues as early as 1 dpi. A clear increase was observed by 10 dpi for ANDV, peaking at 21 dpi. Similarly to ANDV, SNV antigen was already detected in the majority of xenografts at 1 dpi and increased in abundance until 21 dpi. Only occasionally, virus antigen was found at 1 and 3 dpi in HTNV infected xenografts. However, all evaluated xenografts with HTNV infection contained high numbers of virus antigen-positive cells at 10 and 21 dpi, with a peak at 10 dpi. SEOV antigen could be detected at all time points and increased over time, but was found to lower levels compared to ANDV, SNV and HTNV. The number of virus antigen-positive cells in PUUV-infected xenografts was comparatively low and did not show a clear increase over time (S2d Fig). Additionally, given the detection of virus in the murine lungs, virus antigen detection and quantification were also performed in murine lungs and virus antigen was detected in mice inoculated with ANDV, SNV and HTNV (S7a and S7b Fig). Importantly, for all three viruses, virus antigen was more prevalent in murine lungs of mice that were directly inoculated into the human lung xenografts compared to those of non-grafted control mice (S7c Fig). Virus antigen was not detected in murine lungs of SEOV- or PUUV-inoculated mice during the entirety of the study (S7c Fig).

Histopathological analyses demonstrated that tissues with a high number of orthohantavirus antigen-positive cells in general did not show overt lesions. Infected human lung xenografts that contained marked numbers of virus antigen-positive cells presented with only a moderate amount of congestion and no convincing indications of alveolar and interstitial edema or hemorrhage (all score 0). Therefore, no direct correlation was noted between the presence of large numbers of positive cells and significant vascular lesions in histopathology.

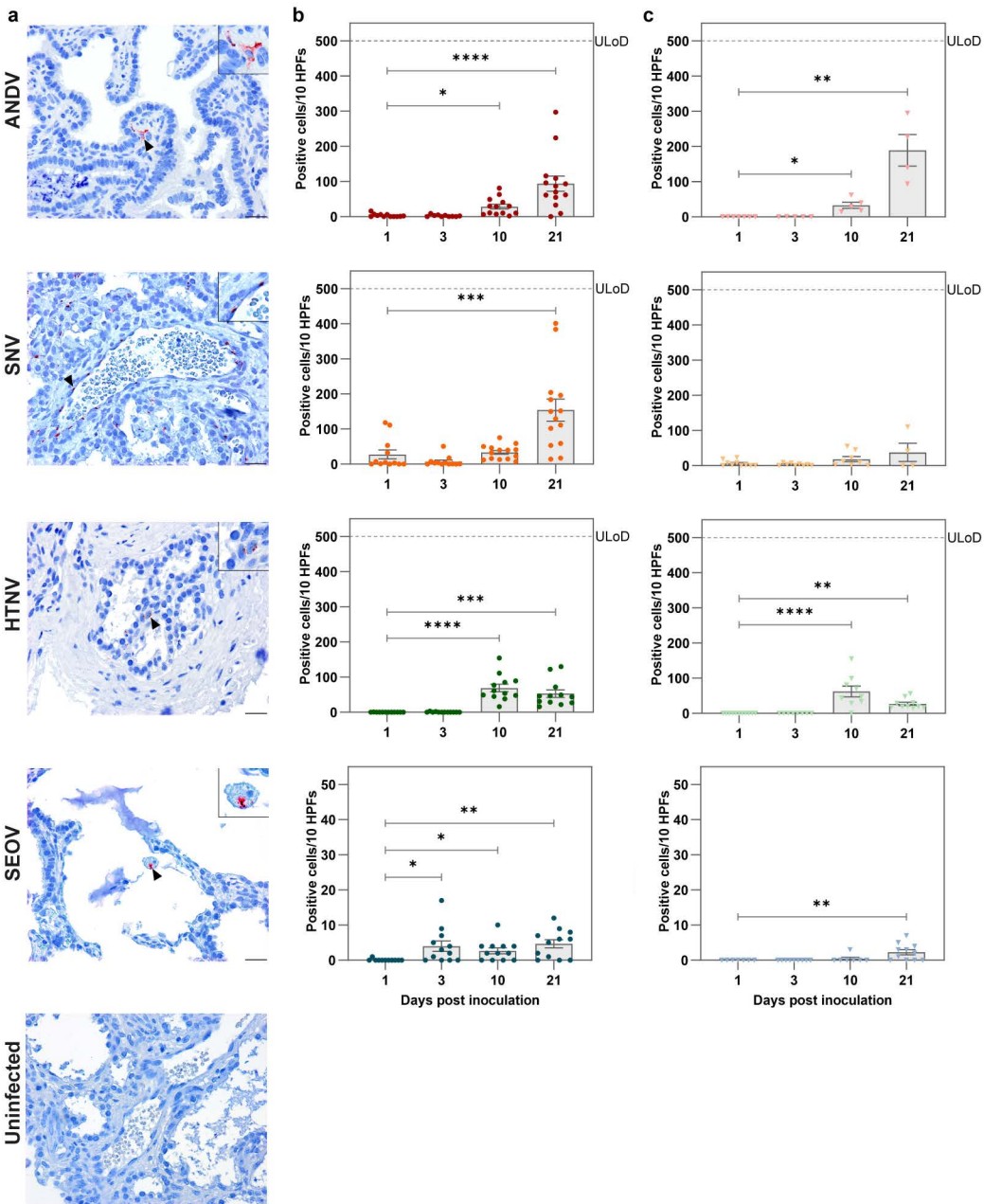

**Fig 4. Quantification of orthohantavirus antigen in human lung xenografts. a)** Orthohantavirus nucleoprotein (N) was detected via immunohistochemistry. Representative images are shown for ANDV-, SNV-, HTNV- and SEOV-inoculated human lung xenografts at 21 days post inoculation (dpi), together with a representative image of human lung xenografts from mice that were left uninfected. Scale bars represent 10 μm. Top right inset images offer a zoom-in of individually infected cells as indicated by arrow heads. **b)** Quantification of virus antigen-positive cells was performed by counting the number of positive cells for antigen staining per ten high power fields (HPFs). Each circle represents one directly inoculated xenograft. **c)** Each light colored triangle represents one non-inoculated xenograft. Bars represent the mean and error bars represent the standard error of the mean. The dashed line indicates upper limit of detection (ULoD). Number of virus antigen-positive cells in directly inoculated (b) and non-inoculated (c) xenografts were compared on 3, 10 and 21 dpi to the number of virus antigen-positive cells on 1 dpi by Kruskall-Wallis test with Dunn's multiple comparisons test. *p < 0.05, **p < 0.005, ***p < 0.001, ****p < 0.0001. Six animals were included per virus per time point.

To identify which cell types in the human lung xenografts were susceptible for orthohantavirus infection, virus antigen was immunofluorescently co-stained with cell markers. Virus antigen was predominantly detected in endothelial cells (Von Willebrand factor⁺), respiratory epithelial cells (cytokeratin⁺) and occasionally macrophages (CD68⁺) (Fig 5). These cell types were consistently infected throughout the study by all tested orthohantaviruses. In conclusion, orthohantaviruses infected primarily endothelium, respiratory epithelium and occasionally macrophages, where infection did not lead to major histopathological changes in the human lung xenografts.

## Local chemo- and cytokine responses in human lung xenografts after orthohantavirus infection

Due to the lack of major virus-associated histopathological lesions, it became of interest to elucidate innate host responses following orthohantavirus infection in the human lung tissues. Therefore, the expression of several human cytokines and chemokines was monitored. Most notably, significantly higher expressions of IP-10 coincided with the peaks of virus antigen presence for all tested orthohantaviruses (Figs 6 and S2e). Furthermore, other markers of orthohantavirus infection of endothelium, respiratory epithelium and macrophages could be

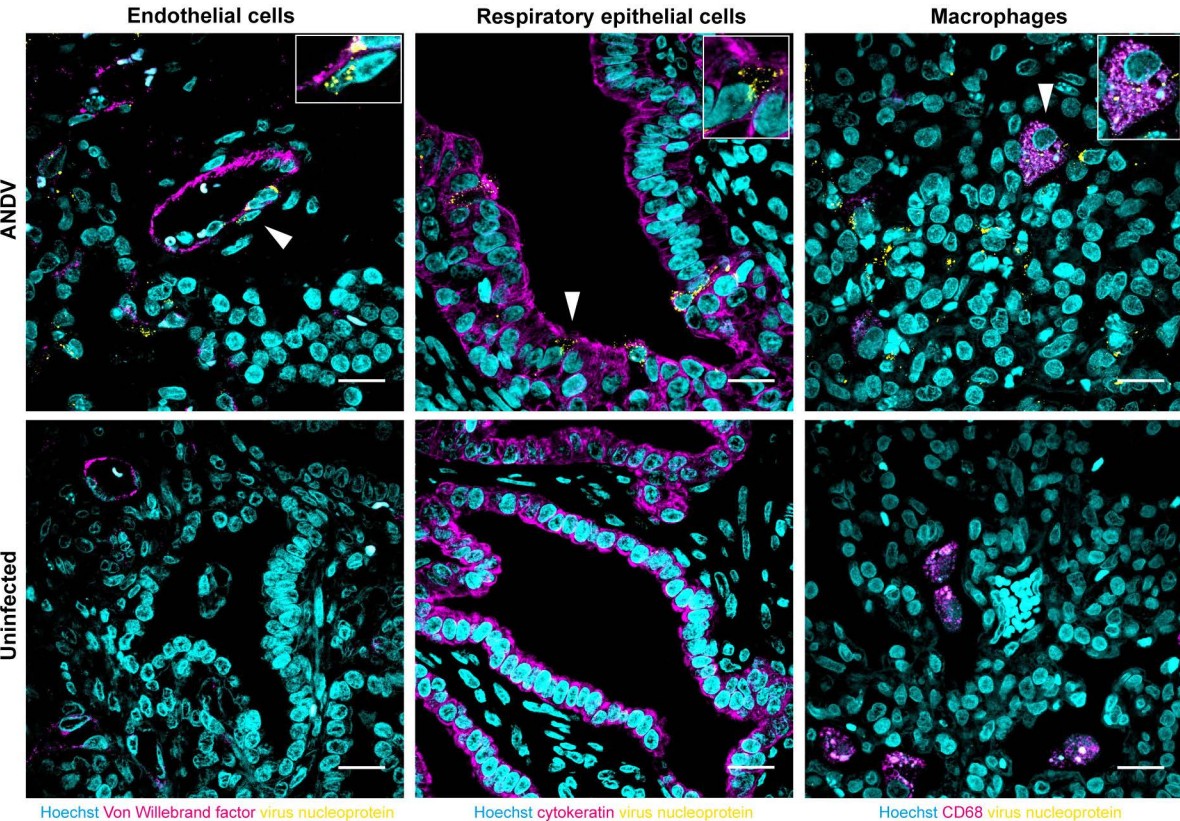

**Fig 5. Orthohantavirus tropism in human lung xenografts.** Representative images for immunofluorescence staining of orthohantavirus antigen in different cell types at 21 days post inoculation. As representative example, ANDV-inoculated human lung xenografts and xenografts from mice that were left uninfected are displayed. Viral tropism for endothelial cells was determined by co-localization of Von Willebrand factor (magenta) and orthohantavirus N (yellow); tropism for respiratory epithelial cells by cytokeratin (magenta) and orthohantavirus N (yellow); tropism for macrophages by co-localization of CD68 (magenta) and orthohantavirus N (yellow). Nuclei are displayed in cyan. Top right inset images offer a zoom-in of individually infected cells as indicated by white arrow heads. Scale bars represent 20 μm.

observed at the peaks of viral replication with increased expression of RANTES by ANDV, SNV and SEOV, increased expression of VCAM-1 by SNV and increased levels of CCL2 by HTNV infection (S8 Fig). Ultimately, while SEOV and PUUV both caused increased levels of IL18, SEOV also caused increased levels of galectin-3 BP, gp130 and IL-15. IFN-gamma and IFN-lambda 2 could not be detected in any analyzed tissues. Taken together, orthohantavirus infection in the human lung xenografts induced a limited innate response that included key inflammatory markers such as IP–10 that could contribute to early events of pathogenesis in the human lung.

## Applicability of the human lung xenograft mouse model for the preclinical evaluation of monoclonal antibodies

To determine if the pan-orthohantavirus model can also be utilized as a preclinical platform to test novel treatment options, we tested a previously described neutralizing monoclonal antibody against ANDV in our model. The monoclonal antibody KL-AN-5E8 neutralizes ANDV via an epitope on the Gn part of the glycoprotein complex and protects Syrian

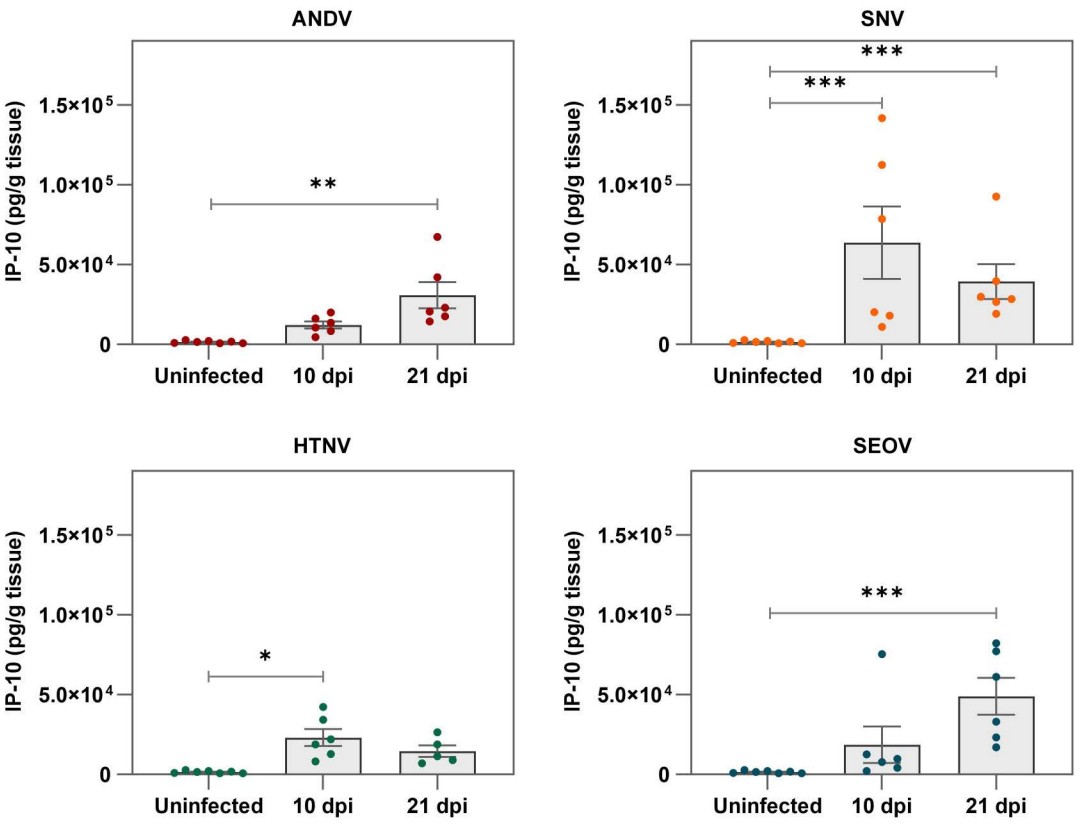

**Fig 6. IP-10 levels in orthohantavirus-infected human lung xenografts.** Human IP-10 levels were measured in tissue homogenates of directly inoculated human lung xenografts. Circles represent individual xenografts, bars represent the mean and error bars represent the standard error of the mean. IP-10 levels were expressed as picogram cytokine per gram tissue. IP-10 levels of orthohantavirus-infected xenografts on 10 and 21 days post inoculation (dpi) were compared to samples from mice that were left uninfected by Kruskall-Wallis test with Dunn's multiple comparisons test. *p < 0.05, **p < 0.005, ***p < 0.001. For each virus group, one xenograft was included per animal per time point (N = 6), with N = 5 for HTNV at 21 dpi. For the uninfected control group, all xenografts from uninfected animals (N = 4) were included.

hamsters against lethal disease [8]. First, the capacity of KL-AN-5E8 to neutralize ANDV and the lack of ANDV neutralization by an irrelevant control (sham) antibody targeting a conserved epitope in the hemagglutinin stem of influenza A and B virus [44] were confirmed *in vitro* ($MNT_{95}$ ranging 1-10 µg/ml for KL-AN-5E8 and $MNT_{95}$ out of detection limit, i.e., >100 µg/ml for the sham antibody). Subsequently, these monoclonal antibodies were administered with the same dose and route as previously in the Syrian hamster model, i.e., 25 mg/kg intraperitoneal [8], one day prior to inoculation and a second dose at 5 dpi (Fig 7a). KL-AN-5E8 was fully protective against viral spread to the non-inoculated human lung xenografts with no detectable ANDV RNA at both 3 and 10 dpi (Fig 7b). Additionally, no viral RNA was found in the sera of KL-AN-5E8-treated animals at both 3 and 10 dpi, demonstrating a complete inhibition of viremia and systemic spread of ANDV. In contrast, the sham antibody did not protect against dissemination of viral RNA into human lung xenografts on both 3 and 10 dpi, and viremia was also detected. This striking difference was less obvious, but also still present in the directly inoculated xenografts as viral RNA loads were significantly reduced in the KL-AN-5E8-treated animals (Fig 7c). At 3 dpi, ANDV RNA loads were 5.43 $Log_{10}$ $TCID_{50}$eq./g (mean; range 0–6.96 $Log_{10}$ $TCID_{50}$eq./g) for KL-AN-5E8-treated animals versus 7.15 $Log_{10}$ $TCID_{50}$eq./g (mean; range 6.08 $Log_{10}$–7.90 $Log_{10}$ $TCID_{50}$eq./g) for sham antibody-treated animals. At 10 dpi, viral RNA loads for KL-AN-5E8-treated animals were lower at 3.81 $Log_{10}$ $TCID_{50}$eq./g (mean; range 0–7.09 $TCID_{50}$eq./g), while these loads were 7.42 $Log_{10}$ $TCID_{50}$eq./g (mean; range 6.06 $Log_{10}$–8.43 $Log_{10}$ $TCID_{50}$eq./g) in sham antibody-treated animals. However, the almost complete absence of virus antigen at 3 and 10 dpi in all examined human lung xenografts of mice treated with KL-AN-5E8 indicated that the detected ANDV RNA in these samples was remaining inoculum (Fig 7d and 7e). Altogether, these data further strengthen the applicability of the current animal model to test protection against infection and systemic spread of orthohantaviruses by neutralizing monoclonal antibodies, such as KL-AN-5E8. In conclusion, this pan-orthohantavirus animal model represents a novel platform to test therapeutics against a broad range of orthohantaviruses.

## Discussion

Orthohantaviruses can cause HCPS and HFRS in humans following inhalation of aerosolized fomites from infected rodents. Although it is commonly described that orthohantaviruses target endothelial cells throughout different organs in the body [4], the early events in the human lungs following inhalation of these viruses are less well understood. There is currently no small animal model that both recapitulates the initial cell tropism and host responses in human lungs, and allows for testing of the efficacy of therapeutics against multiple orthohantaviruses in the same animal. Therefore, we established a human lung xenograft model in NSG mice and tested its permissiveness for distinct orthohantaviruses. As previously described, maturated human lung tissues developed structures resembling bronchioles lined with respiratory epithelium, cartilage and alveolar-like spaces [35]. These human lung tissues allowed for replication of pathogenic orthohantaviruses such as HCPS-associated ANDV and SNV, as well as HFRS-associated HTNV, SEOV and the low pathogenic PUUV. This highlights the utility as a pan-orthohantavirus animal model while also representing a crucial platform in outbreak preparedness against novel emerging orthohantavirus species, therefore being unique when compared to existing animal models to date. Although replication of the orthohantaviruses in this study did not result in associated histopathological changes in human tissue or overt clinical signs, the demonstrated breadth of permissiveness of the model for distinct orthohantaviruses offers an exclusive preclinical platform for new cross-protective pan-orthohantavirus intervention strategies.

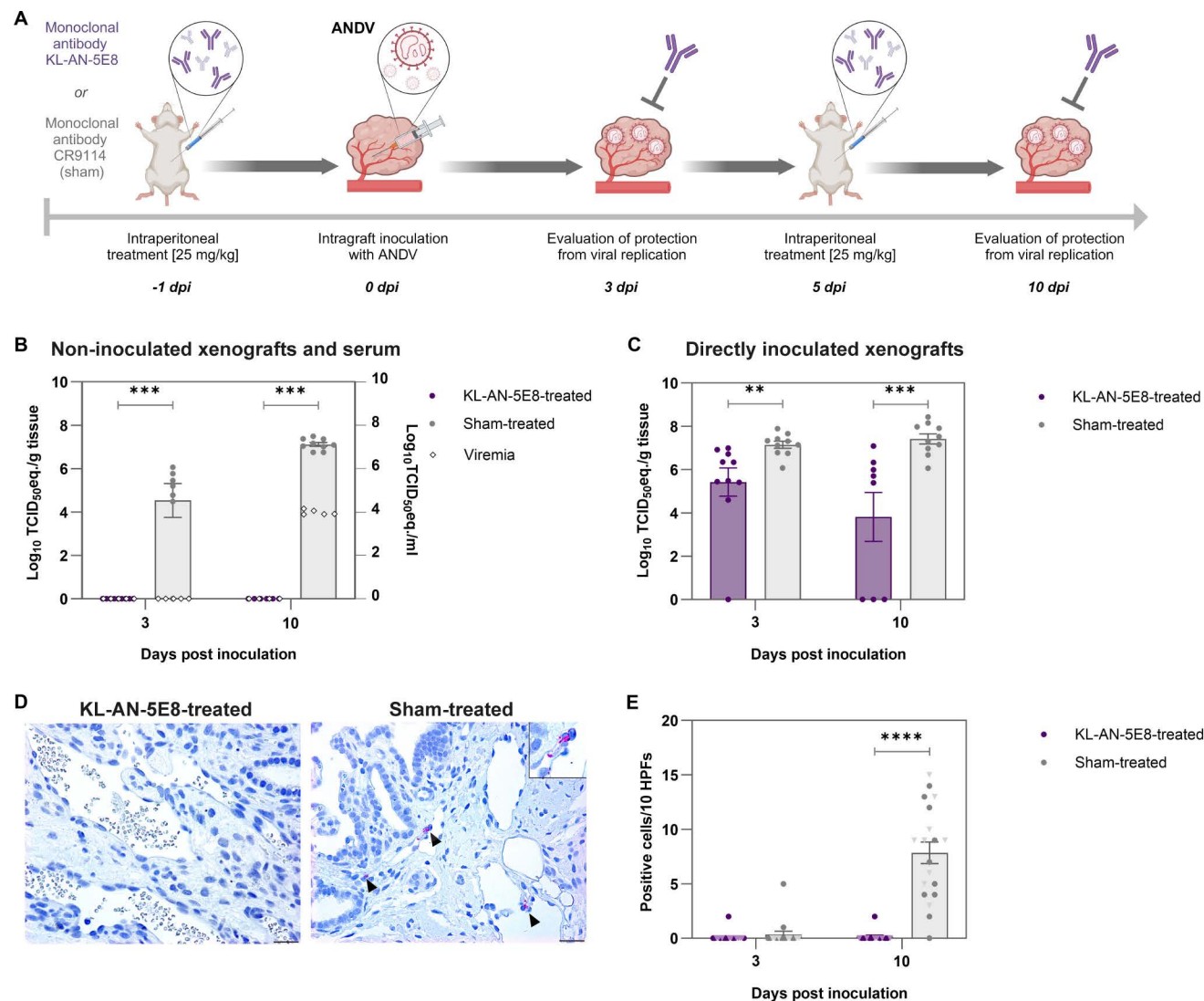

**Fig 7. Preclinical application of the human lung xenograft mouse model. a)** Overview of the timeline and experimental set-up of the preclinical antibody prophylactic protection study. Mice were either treated with 25 mg/kg monoclonal antibody KL-AN-5E8 or CR9114 (sham) intraperitoneally one day prior to infection. Subsequently, mice were infected via intragraft inoculation of 30 µl of $10^6$ TCID$_{50}$/ml ANDV in maximum two human lung xenografts. Remaining xenografts were left non-inoculated. Mice treated with KL-AN-5E8 (N = 5) or sham antibody (N = 5) were necropsied at 3 days post inoculation (dpi). The remaining mice received an additional dose of 25 mg/kg monoclonal antibody. At 10 dpi, the remaining mice (N = 4 for KL-AN-5E8; N = 5 for sham) were necropsied and evaluated for protection against viral replication. *Created in BioRender. 1, V. (2025)* https://BioRender.com/m32p764. **b)** ANDV RNA loads were quantified by RT-qPCR in homogenates of non-inoculated human lung xenografts and in murine serum. Each circle indicates one evaluated xenograft, each unfilled diamond represents viremia of one single animal. **c)** ANDV RNA loads were quantified in homogenates of directly inoculated human lung xenografts. Viral RNA loads were expressed as TCID$_{50}$ equivalents per gram tissue on the left Y-axis, or TCID$_{50}$ equivalents per ml serum on the right Y-axis. Error bars represent the standard error of the mean. **d)** ANDV N was detected via immunohistochemistry. Representative images of human lung xenografts are shown for KL-AN-5E8- and sham-treated mice at 10 dpi. Presence of virus antigen is indicated by arrow heads. Scale bars represent 10 µm. Top right inset image offers a zoom-in of individually infected cells. **e)** Quantification of virus antigen-positive cells was performed by counting the number of positive cells for antigen staining per ten high power fields (HPFs). Each symbol indicates one evaluated xenograft. Circles represent directly inoculated xenografts and light colored triangles represent non-inoculated xenografts. Bars represent the mean and error bars represent the standard error of the mean. Results of each treatment group for viral RNA loads in non-inoculated (b), directly inoculated (c) xenografts, and number of viral antigen-positive cells in all analyzed xenografts (e) were compared with a two-tailed Mann Whitney U test on 3 and 10 dpi. **p < 0.005, ***p < 0.001, ****p < 0.0001.

The infected cell types in the current model represent those described for lethal HCPS infection in humans and a limited number of HFRS-associated PUUV cases, which include pulmonary endothelial cells, epithelial cells and macrophages [2,3,45,46]. Infection of these cells results in increased expression of key inflammatory mediators IP-10 and RANTES by multiple orthohantaviruses, as observed previously *in vitro* [47] and are also in line with the observed upregulation of genes encoding IP-10 (*CXCL10*) and RANTES (*CCL5*) in ANDV-inoculated hamster models that recapitulate human HCPS symptoms [34]. IP-10 can be expressed by different cell types, like endothelial cells, and can act as chemoattractant for macrophages, T cells, natural killer cells and dendritic cells, which all have been described to play an important role in orthohantavirus pathogenesis [2,48,49]. Similar as in our model, IP-10 is increased in human HCPS [50,51] and HFRS patients [50–52], however, not in HTNV-inoculated hamsters, which do not demonstrate significant increased expression of *CXCL10* [34]. Increased expression of these markers indicates that infection of these cells triggers next steps in pathogenesis which could involve a Th1 immune response by attracting mononuclear leukocytes such as activated monocytes and T cells, potentially leading to increased vascular permeability as observed in HCPS and HFRS cases [47]. Also, one of the small number of *in vitro* studies that investigated ANDV infection in human lung cells using a co-culture model of human bronchial epithelial cells and a lung fibroblast cell line, demonstrated increases in pro-inflammatory markers such as IP-10, but also IL-6 and IL-8 [53]. Whereas host responses in human lung xenografts to orthohantavirus infection were consistently associated with increased levels of human IP-10, no significant increase of IL-6, a described marker of disease severity [54,55] or IL-8, were detected in our study. An explanation for this can be the limited abundance of cytokine producing immune cells and the relatively low total number of infected cells in respective human lung xenografts, limiting the detection of inflammatory cytokines to those that are substantially increased, also in tissue with limited viral replication. Additionally, the lack of a functional human immune system in these mice is a likely explanation for the lack of histopathological lesions post infection in this human lung-only mouse model. Previous studies demonstrated a crucial role for neutrophils and (adoptively transferred) T cells in HTNV-induced pulmonary edema in SCID mice [56] and for CD8[+] T cells in HTNV-induced pathogenesis in NSG mice with a humanized immune system [57]. As previously demonstrated [37,58], a fully functional human immune system can be reconstituted in the current mouse model to specifically study the implied hypothesis of induced immunopathogenesis, offering a great future tool for further characterization of orthohantavirus pathogenesis.

More efforts are being made to replace, reduce and refine animal experimentation by generating novel complex cell culture models that represent human organs. Although these (organoid-derived) respiratory epithelial air-liquid interface cultures are proving to be valuable tools for the study of respiratory viruses and testing of novel therapeutics, they also present limitations compared to our current model. Firstly, these cultures only contain a limited number of distinct respiratory epithelial cell types, and generally lack cell types such as mesenchymal cells, fibroblasts, and more importantly, targets for orthohantavirus infection such as endothelial cells and macrophages. Secondly, these models do not allow for the study of the potential of therapeutics to protect against viral dissemination through systemic blood circulation. However, in our current model the xenografts do not represent fully functional lungs due to the lack of air-organ interface, which limits the translational potential of the model for the study of initial steps of infection including aerosol transmission. Nevertheless, although the intragraft inoculation route might present a less physiologically translational route, the additional inclusion of multiple non-inoculated human lung xenografts per mouse facilitates monitoring of viral spread from the vasculature to the human respiratory tissues and vice versa. Additional advantage of engraftment of the human lung tissues is that it

facilitates more reproducible and efficient replication for multiple orthohantaviruses when compared to non-grafted NSG mice in this study. Also, as observed during the current study, this mouse model allows for viral spread throughout the murine circulation and different organs. This renders the current model more reproducible and suitable to assess the efficacy of direct countermeasures, such as neutralizing antibodies, that inhibit systemic viral spread, as compared to mouse models based on immunocompetent mice that do not allow for detection of viral antigen throughout the mouse [59]. However, the even more important advantage is that the current set-up of the model allows for the study of early events during infection in actual human lung tissue.

Although our current model does not recapitulate disease as observed in humans, recovery of infectious virus, viral RNA, virus antigen in the human lung tissue, viremia and expression levels of IP-10 could serve as proxy markers for orthohantavirus infection and can therefore be used as readout for preclinical studies. Future studies using the fully humanized lung xenograft model will allow us to study the role of the inflammatory response in the pathogenesis of different orthohantavirus species in the human lungs.

In summary, we established a novel *in vivo* pan-orthohantavirus infection model that allows for viral replication in highly relevant human lung tissue. A neutralizing monoclonal antibody was used as a proof-of-concept to demonstrate that this model can be utilized as a preclinical platform to test novel therapeutics targeting known and emerging orthohantaviruses. This novel animal model, in particular with the perspective of human immune cell reconstitution, will be an invaluable tool for investigating the comparative virulence of orthohantaviruses with unclear pathogenic potential in the human lung, and for testing the broad reactivity of novel candidate pan-orthohantavirus antivirals and antibodies.

## Methods

### Ethics statement

Research involving animals was conducted in compliance with the Dutch legislation for the protection of animals used for scientific purposes (2014, implementing EU Directive 2010/63) and other relevant regulations. The licensed establishment where this research was conducted (Erasmus MC) has an approved Office of Laboratory Animal Welfare (OLAW) Assurance # F16-00046 (A5051-01). Research was conducted under a breeding protocol (#122-58-01) and project license (#199106) from the Dutch competent authority and the study protocol was approved by the institutional Animal Welfare Body. Human fetal lung tissue was obtained from legally terminated second trimester pregnancies (15–20 weeks) by the Human Immune System-Mouse Facility of Academic Medical Center (AMC; Amsterdam, the Netherlands), after written informed consent of the mother for the tissue's use in research and with approval of the Medical Ethical Review Board of the AMC (MEC: 03/038). Study procedures were performed according to the Declaration of Helsinki, and in compliance with relevant Dutch laws and institutional guidelines. The tissues obtained were anonymized and non-traceable to the donor and only the gestational age was provided.

### Animals

During the breeding period as well as post-surgery, animals were housed in individually ventilated cages (IVC green line, Tecniplast) in groups of maximum 4 animals. After inoculation with orthohantaviruses, animals were housed in groups of maximum 4 animals in filter top cages (type 1, Tecniplast), in Class III isolators, allowing social interactions, under controlled conditions of humidity, temperature and light (12-hour light/12-hour dark cycles). Irradiated food and acidified water were available *ad libitum*. Animals were cared for and monitored

 

(pre and post inoculation) daily by qualified personnel. All animals were allowed to acclimatize to husbandry for at least 7 days. For unbiased experiments, all animals were randomly assigned to experimental groups. The animals were anesthetized (2% isoflurane) for all invasive procedures. Mice were euthanized by cardiac puncture and exsanguination.

## Viruses and cells

Andes orthohantavirus strain 9717869 (ANDV, European Virus Archive Global #002v-EVA400, P+2), Sin Nombre orthohantavirus strain Convict Creek 107 (SNV, #008v-EVA401, P+2), Hantaan orthohantavirus strain 76-118 (HTNV, #008v-EVA1471, P+3), Seoul orthohantavirus strain 80-39 (SEOV, #008v-EVA1473, P+2) and Puumala orthohantavirus strain Cg 18-20 (PUUV, #007v-00809, P+3) stocks were propagated on Vero E6 cells (CRL-1586, ATCC, United States) in Dulbecco's modified Eagle's medium (DMEM, Capricorn Scientific, Ebsdorfergrund, Germany) supplemented with 5% fetal calf serum (FCS), HEPES, sodium bicarbonate, 100 IU/ml penicillin and 100 µg/ml streptomycin (pen/strep) (Lonza, Basel, Switzerland) at 37°C in a humidified $CO_2$ incubator (S1 Table). Vero E6 cells were cultured in DMEM supplemented with 10% FCS, HEPES, sodium bicarbonate and pen/strep at 37°C in a humidified $CO_2$ incubator. All cells and virus stocks were confirmed to be free of mycoplasma.

## Experimental infection of human lung xenografted mice with orthohantaviruses

Breeding pairs of NSG mice (NOD.Cg-Prkdc$^{scid}$Il2rg$^{tm1Wjl}$/SzJ) were kindly provided by Rosalie Joosten (Laboratory for Stem Cell and Cancer Research, Erasmus MC, Rotterdam, the Netherlands) or purchased (Jackson Laboratory, Maine, United States). Animals for experimental purposes were bred at the Erasmus MC. Female and male mice (at least 8 weeks old) underwent xenografting by subcutaneous implantation of fetal human lung tissues. For this, three pieces each (approximately 3-6 mm$^3$) of human fetal lung tissue were implanted into 4 separate subcutaneous pockets onto the back of mice under inhalation anesthesia (2-3% isoflurane) and incisions were sutured (Vicryl 6-0, Ethicon, New Jersey, United States). All mice received a pre-surgical antibiotic supply via the drinking water (one day pre-surgery) and analgesic treatment (Carprofen, Carporal, AST pharma, Oudewater, the Netherlands). Engraftment was monitored visually and by palpation. After a minimum maturation period of 12 weeks, animals were transferred to the animal biosafety level 3 (ABSL-3) biocontainment laboratory and were given an acclimatization time of at least 7 days. Corresponding to their designated experimental group, animals were infected by an intragraft inoculation of 30 µl of 1.0 x 10$^6$ median tissue culture infectious dose (TCID$_{50}$) of ANDV, SNV, HTNV, SEOV and PUUV (S2 Table). An initial pilot experiment with PUUV was performed (N = 4 per time point) where all available human lung xenografts were directly inoculated. After replication of PUUV was confirmed, the model was used to study the permissiveness for ANDV, SNV, HTNV and SEOV. Therefore, in the following experiments, a maximum of two human lung xenografts were directly inoculated, whereas all remaining xenografts (maximum 2) were left non-inoculated to study the potential of systemic viral spread. On 1, 3, 10 and 21 dpi, animals (N = 6) from each experimental group were euthanized. Non-grafted animals were subcutaneously inoculated with 30 µl of 1.0 x 10$^6$ TCID$_{50}$ as controls for susceptibility of the NSG mice to orthohantavirus infection. Grafted, but non-inoculated animals served as control for human lung tissue composition. Both groups of control animals were sacrificed at 21 dpi. Following infection, the body weight of all animals was monitored regularly. During all necropsies, the xenografts, murine lungs, liver, kidney, spleen and serum were collected for quantification of infectious virus

and viral RNA load. Organs were stored in 4% neutral-buffered formalin, embedded in paraffin, sectioned at 3 μm, and stained with hematoxylin and eosin (H&E) for histopathology and processed for immunohistochemistry (IHC).

## Application of the lung xenograft mouse model for preclinical evaluations

Xenografted mice (S3 Table) were treated with 25 mg/kg (intraperitoneal) of a neutralizing mouse monoclonal antibody targeting the Gn component of the glycoprotein complex of ANDV, KL-AN-5E8 [8], or were sham-treated with an irrelevant neutralizing human monoclonal antibody targeting the hemagglutinin stem of influenza A and B viruses, CR9114 [44]. One day after initial treatment, all animals were challenged with ANDV via intragraft inoculation (30 μl, 1.0 x $10^6$ $TCID_{50}$ of ANDV into a maximum of two xenografts). Mice were treated with a second dose of antibody at 5 dpi, following the same administration schedule as described above. The body weight of all animals was monitored regularly. On 3 and 10 dpi, representing time points of initial replication and the beginning of plateaued viral replication, animals from each experimental group (S3 Table) were euthanized. The human lung xenografts, as well as the murine lungs, liver, kidney, spleen and serum were collected for quantification of viral RNA load. Xenografts were analyzed for histopathology and presence of virus antigen.

## RNA isolation and virus titer quantification by RT-qPCR

To determine presence of the viral RNA load by reverse transcription quantitative PCR (RT-qPCR) and infectious viral titers by $TCID_{50}$ assay, small pieces of tissues were collected, weighed and homogenized. Additionally, serum samples were collected. Prior to aliquoting, all samples were centrifuged at 4,000 g for 10 min. The first aliquot was prepared for RNA isolation by collecting 60 μl of each homogenate in 90 μl MagNA Pure 96 External Lysis Buffer (Roche, Basel, Switzerland). A second aliquot of 60 μl was collected for multiplex bead-assay, after which a final aliquot of the samples was stored at -80°C until virus titration. Prior to RT-qPCR, a known concentration of phocine distemper virus (PDV) was added to each sample as internal control for RNA extraction [60]. RNA extraction was performed as described previously with the modification that RNA was eluted in 30 μl PCR grade water [61]. RT-qPCR was performed utilizing a 7500 Real-Time PCR System (Applied Biosystems) with Taqman primer-probe mixes targeting the S segment of different orthohantaviruses (S4 Table). Ct-values were converted to $TCID_{50}$ equivalents per gram tissue or ml ($TCID_{50}$eq./g or $TCID_{50}$eq./ml) by comparison to standard curves derived from corresponding orthohantavirus stocks titrated on Vero E6 cells that were included during each round of RT-qPCR.

## Virus titer quantification with endpoint dilution assay

Tissue homogenates were cleared by centrifugation at 4,000 g for 10 minutes. Infectious virus titers were quantified by $TCID_{50}$ dilution assay. Ten-fold serial dilutions of homogenates were added in triplicate to monolayers of Vero E6 cells. Cells were incubated for 5 days at 37°C. Supernatants were removed and cells were fixed with ice-cold absolute ethanol for 30 minutes at −20°C. Then, cells were incubated with 70% ethanol for 5 minutes at room temperature and washed three times with phosphate buffered saline (PBS). Cells were blocked in 5% dried milk powder (Campina, Zaltbommel, the Netherlands) diluted in PBS for 1 hour at room temperature. As the mouse monoclonal antibody TULV 1 efficiently binds orthohantavirus nucleoprotein (N) from all viruses utilized in this study, except for PUUV [62], this antibody was used as primary antibody to detect orthohantavirus N, 1:50 diluted in 1% milk in PBS overnight

at 4°C. Staining against PUUV N was performed with a mouse monoclonal antibody 5E11 (1:100, Abcam, Cambridge, United Kingdom) as primary antibody. After three washes with PBS, cells were stained with Alexa Fluor 488-labeled donkey anti-mouse IgG antibody (Invitrogen, Waltham, United States) 1:1000 diluted in 1% milk in PBS. After three washes with PBS, cells were stored in PBS at 4°C until detection of orthohantavirus N-positive cells was performed using an Axio Observer Vert. 1A fluorescence microscope (Zeiss, Oberkochen, Germany). TCID$_{50}$/ml values were calculated according to methods described by Spearman & Kärber.

## Histopathology and immunohistochemistry

To detect presence of virus antigen by IHC and to perform histopathological examinations, the human lung xenografts and murine organs were fixed in 4% neutral-buffered formalin for two weeks, after which these tissues were embedded in paraffin and sectioned at 3 μm. To evaluate histopathological changes, sections were stained with H&E. Virus antigen expression was detected by IHC of all human and murine lung tissues. Paraffin was removed from sections, which were rehydrated and pretreated with citric acid buffer (pH 6.0) for 15 minutes at 100°C and washed with PBS. Endogenous peroxidase was blocked with 3% hydrogen peroxide for 10 minutes. Slides were washed with Milli-Q water and then blocked with 10% normal rabbit serum (DAKO, Santa Clara, United States) for 30 minutes at room temperature. Slides were then incubated with either TULV 1 (1:50) or 5E11 (1:100; for PUUV) in 1% bovine serum albumin (BSA, Sigma, St. Louis, United States) in PBS for overnight incubation at 4°C. After washing three times with 0.05% PBS-Tween 20, slides were incubated with horseradish peroxidase (HRP) labeled rabbit anti-mouse IgG (1:100, DAKO, Santa Clara, United States) diluted in 1% BSA in PBS for 1 hour at room temperature. Subsequently, slides were washed three times with PBS and HRP activity was revealed by incubating slides in 3-amino-9-ethylcarbazole (Sigma, St. Louis, United States) solution for 10 minutes. Ultimately, sections were counterstained with hematoxylin and embedded with Kaiser's glycerol gelatin. Immunoreactivity for N protein of different orthohantaviruses was shown as red granular intracytoplasmic staining. Paraffin-embedded samples of Vero E6 cells, which were previously infected with different orthohantaviruses were utilized as positive controls. Tissue sections from uninfected mice were utilized as negative controls. In parallel, different cell populations and potential orthohantavirus receptors were stained in human lung xenograft sections from uninfected mice. Endothelial cells were stained with a mouse monoclonal anti-Von Willebrand factor antibody (1:50, clone F8/86, Invitrogen, Waltham, United States), epithelial cells with a mouse monoclonal anti-cytokeratin 19 antibody (1:250, clone A53-B/A2, Abcam, Cambridge, United Kingdom), immune cells with a mouse monoclonal anti-CD45 antibody (1:50, clone 2B11+PD7/26, Agilent, Santa Clara, United States), protocadherin-1 with a mouse monoclonal anti-protocadherin-1 antibody (1:25, clone B-11, Santa Cruz Biotechnology, Dallas, United States), and CD55 with a mouse monoclonal anti-CD55 antibody (1:25, clone NaM16-4D3, Santa Cruz Biotechnology, Dallas, United States). As negative controls, corresponding isotype control antibodies in corresponding dilutions were utilized with mouse IgG1 isotype control antibody (RnD Systems, Minneapolis, United States) for endothelial cells, immune cells, protocadherin-1 and CD55, and mouse IgG2a isotype control antibody (RnD Systems, Minneapolis, United States) for epithelial cells. For β3 integrin staining, samples were blocked with 10% normal goat serum (DAKO, Santa Clara, United States), primary antibody was a rabbit monoclonal anti-β3 integrin antibody (1:50, clone SJ19-09, Novus Biologicals, Abingdon, United Kingdom) followed by secondary HRP labeled goat anti-rabbit IgG antibody (1:100, DAKO, Santa Clara, United States). Rabbit IgG antibody (RnD Systems, Minneapolis, United States) was used as isotype control.

## Histopathological analyses

Immunohistochemical scoring was evaluated by counting the number of orthohantavirus antigen-positive cells in 10 random 400x high power fields (HPFs) consisting of at least 80% respiratory tissue. Whenever a HPF did not contain sufficient respiratory tissue, i.e., presence of large amounts of cartilage or presence of large empty lumen, a new HPF was selected until 10 HPFs were evaluated. If a single xenograft was not large enough to contain 10 400x HPFs, the maximum achievable number of HPFs was taken and thereafter HPFs were selected randomly again to reach the total of 10. Additionally, the cell types positive for virus antigen were evaluated. A basic histopathological scoring for lesions was conducted. The presence and number of extravasated erythrocytes in the alveolar lumina (alveolar hemorrhages) and in the interstitium (interstitial hemorrhages) of both the human lung xenografts and the murine lungs were evaluated on a scale from 0 to 4 based on the lesion severity grade (hemorrhage; 0: none, 1: slight, 2: mild, 3: moderate, 4: severe). In a similar manner the presence and severity of edema was assessed on the same scale of 0 to 4 (edema; 0: none, 1: slight, 2: mild, 3: moderate, 4: severe) characterized by loosely arranged matrices in dilated optic empty spaces and/or presence of lightly eosinophilic reticular material.

## Immunofluorescence staining

Paraffin-embedded tissue slides were prepared and antigen retrieval was performed as described for IHC. For immunofluorescence, slides were blocked for 30 minutes at room temperature with 10% normal goat serum (DAKO, Santa Clara, United States). Subsequently, slides were incubated with dilutions of primary antibody in 1% BSA in PBS. Co-staining was performed for the evaluation of viral tropism. Virus antigen was detected with TULV 1 antibody (1:50), combined with the anti-Von Willebrand factor antibody (1:50) as endothelial cell marker, a mouse anti-cytokeratin pan type I/II IgG1 antibody cocktail (1:100, clone AE1/AE3, Invitrogen, Waltham, United States) as respiratory epithelial cell marker and a mouse monoclonal anti-CD68 antibody (1:200, clone KP1, DAKO, Santa Clara, United States) as macrophage marker. Slides from uninfected animals and isotype controls were utilized as negative controls. After overnight incubation of primary antibodies at 4°C, slides were washed 3 times with PBS. Slides were incubated for 1 hour at room temperature with secondary antibodies diluted in 1% BSA in PBS. Cell markers were stained with Alexa Fluor 488-conjugated goat anti-mouse IgG1 secondary antibody (1:250, Invitrogen, Waltham, United States) and virus antigen with Alexa Fluor 594-conjugated goat anti-mouse IgG2a secondary antibody (1:250, Invitrogen, Waltham, United States). After washing twice with PBS, Hoechst 33342 solution (Thermo Scientific, Waltham, United States) was added for 10 minutes before samples were washed again twice with PBS, mounted with ProLong Diamond Antifade Mountant (Invitrogen, Waltham, United States) and stored at 4°C until imaging. Immunofluorescent samples were imaged using a Zeiss LSM700 confocal laser scanning microscope with ZEN software (Zeiss, Oberkochen, Germany). Representative images were acquired and co-staining images were demonstrated as maximum intensity projections of Z-stacks. Color optimizations for visualization were performed in ImageJ software.

## Multiplex bead-assay

Tissue homogenates of human xenograft tissues were tested for human cytokine and chemokine concentrations by a customized Magnetic Luminex assay (RnD Systems, Minneapolis, United States). Concentrations of the following markers were measured using a Bio-Plex 200 system (Bio-Rad, California, United States): interferon gamma-induced protein 10 (IP-10), galectin-3 binding protein (galectin-3 BP), intercellular adhesion molecule (ICAM-1),

interferon-beta (IFN-β), IFN-γ, IFN-λ2, interleukin-6 (IL-6), IL-8, IL-15, IL-18, IL-6 receptor alpha (IL-6R alpha), IL-6R beta, E-selectin, monocyte chemoattractant protein-1 (MCP-1), RANTES, vascular cell adhesion molecule 1 (VCAM-1) and vascular endothelial growth factor (VEGF). Samples were two-fold diluted and run in duplicate. For infected animals, only tissue homogenates positive for viral RNA were included and samples weighing less than 10 mg were excluded from analyses. Values below the limit of detection were replaced by the value of the lower limit of detection to allow for statistical testing. Cytokine levels were standardized per gram tissue and inoculated samples were compared to samples from uninfected animals.

## Data visualization and statistical analyses

Graphs and heatmaps were created using GraphPad Prism 10 software (La Jolla, CA, United States). Adobe Illustrator software was utilized to process the figures into the final layout. All statistical analyses were performed using GraphPad Prism 10 software. No statistical methods were applied to predetermine sample size. Applied statistical tests are indicated in according figure or table captions. In general, Kruskall-Wallis test with Dunn's multiple comparisons test was utilized to compare infectious viral titers, viral RNA loads or number of virus antigen-positive cells at different time points. This test was also applied to compare human cytokine levels at 10 and 21 dpi with uninfected animals. In the preclinical setting, a two-tailed Mann Whitney U test was utilized to compare viral RNA loads and number of virus antigen-positive cells between the two treatment groups. For all statistical comparisons $p < 0.05$ was considered statistically significant.

## Supporting information

**S1 Fig. Replication kinetics of distinct orthohantaviruses by RT-qPCR.** a) Orthohantavirus small (S) segment RNA loads were quantified in human lung xenografts that were directly inoculated with ANDV, SNV, HTNV and SEOV by RT-qPCR. b) Orthohantavirus S segment RNA loads in non-inoculated xenografts were quantified by RT-qPCR. Squares indicate the mean $TCID_{50}$ equivalent per gram tissue and error bars represent the standard error of the mean. RNA loads in directly inoculated xenografts (a) and non-inoculated xenografts (b) were compared to the lowest RNA load during the course of infection, i.e., 1 or 3 days post inoculation (dpi) by Kruskall-Wallis test with Dunn's multiple comparisons test. *$p < 0.05$, **$p < 0.005$, ***$p < 0.001$, ****$p < 0.0001$. Six animals were included per virus per time point. (TIF)

**S2 Fig. Puumala orthohantavirus (PUUV) replication in the human lung xenograft mouse model.** a) Infectious viral titers were quantified in human lung xenografts. Squares indicate the mean $TCID_{50}$ per gram tissue and error bars represent the standard error of the mean. b) PUUV S segment RNA loads were quantified in human lung xenografts with RT-qPCR. No viremia was detected. Squares indicate the mean $TCID_{50}$ equivalent per gram tissue and error bars represent the standard error of the mean. c) PUUV nucleoprotein (N) was detected via immunohistochemistry. A representative image is shown for infection at 21 days post inoculation (dpi). Presence of virus antigen is indicated by an arrow head. Scale bar represents 10 μm. Top right inset image offers a zoom-in of an individually infected cell as indicated by the arrow head. d) Quantification of virus antigen detection was performed by counting the number of positive cells for antigen staining per ten high power fields (HPFs). Each circle indicates one evaluated xenograft. Bars represent the mean and error bars represent the standard error of the mean. Infectious viral titers (a), viral RNA loads (b) and number of virus antigen-positive cells (d) in human lung xenografts were compared on 3, 10 and 21 dpi to

the infectious viral titers (a), RNA loads (b) or number of virus antigen-positive cells (d) on 1 dpi by Kruskall-Wallis test with Dunn's multiple comparisons test. e) Human IP-10 levels were measured in tissue homogenates of PUUV-inoculated human lung xenografts. Each circle represents one evaluated xenograft, bars represent the mean and error bars represent the standard error of the mean. IP-10 levels were expressed as picogram cytokine per gram tissue and infected samples were compared to samples from mice that were left uninfected by Kruskall-Wallis test with Dunn's multiple comparisons test. *p < 0.05.
(TIF)

**S3 Fig. Relative body weight of lung xenografted mice.** Relative body weight of human lung xenografted mice, following ANDV-, SNV-, HTNV-, SEOV- and PUUV-inoculation. Xenografted mice that were left uninfected were included as an additional control group. The mean of each group is shown as a solid line, error bars represent standard of the mean. The dashed line is indicating the humane endpoint for the total loss of body weight.
(TIF)

**S4 Fig. Detection of orthohantaviruses in murine tissues.** a) Infectious viral titers were quantified in murine lungs of ANDV-, SNV-, HTNV- and SEOV-inoculated animals. No infectious virus was detected in murine lungs of PUUV-inoculated animals. Circles indicate the mean $TCID_{50}$ per gram tissue and error bars represent the standard error of the mean. b) Orthohantavirus S segment RNA loads were quantified in murine lungs with RT-qPCR. Circles indicate the mean $TCID_{50}$ equivalent per gram tissue and error bars represent the standard error of the mean. Infectious viral titers (a) and viral RNA loads (b) in murine lungs were compared to the lowest infectious viral titers (a) or RNA loads (b) in murine lungs during the course of infection, i.e., at 1 or 3 dpi by Kruskall-Wallis test with Dunn's multiple comparisons test. *p < 0.05, **p < 0.005, ***p < 0.001. c) Heatmap displaying quantification of orthohantavirus S segment RNA loads in murine lungs, kidney, liver and spleen as determined by RT-qPCR. Each cell represents the mean per group (N = 6, N = 4 for PUUV) expressed in $TCID_{50}$ equivalents per gram tissue. Grey cells represent that all samples within a group were below lower limit of detection.
(TIF)

**S5 Fig. Detection of orthohantavirus RNA loads in non-grafted mice.** a) Heatmap displaying quantification of orthohantavirus S segment RNA loads in murine lungs, kidney, liver, spleen, cranial and caudal skin as determined by RT-qPCR. Each cell represents the value of each individual animal expressed in $TCID_{50}$ equivalents per gram tissue. Grey cells represent values below lower limit of detection. X represents undetermined values. b) Comparison of orthohantavirus S segment RNA loads in murine lungs of xenografted and non-grafted NSG mice. Colored circles indicate the values for murine lungs of human lung xenografted animals, whereas grey circles indicate those of non-grafted animals.
(TIF)

**S6 Fig. Donor- and sex-dependent variations of viral RNA loads in human lung xenografts.** a) Comparison of orthohantavirus S segment RNA loads in directly inoculated human lung xenografts. Results are depicted by human donors and inoculated viruses over time on 1, 3, 10 and 21 days post inoculation and expressed in $TCID_{50}$ equivalents per gram tissue. Every symbol represents a single human lung xenograft. b) Comparison of orthohantavirus S segment RNA loads in directly inoculated human lung xenografts depicted by the sex of mice and inoculated viruses over time on 1, 3, 10 and 21 days post inoculation. Every symbol represents a single human lung xenograft.
(TIF)

**S7 Fig. Quantification of orthohantavirus antigen in murine lungs.** a) Orthohantavirus nucleoprotein (N) was detected via immunohistochemistry. Representative images are shown from the murine lungs of ANDV-, SNV-, and HTNV-inoculated mice at 21 days post inoculation (dpi), together with a representative image of murine lungs from mice that were left uninfected. Scale bars represent 10 μm. Presence of virus antigen is indicated by arrow heads. No virus antigen was detected in the murine lungs of SEOV- and PUUV-inoculated mice. b) Quantification of virus antigen-positive cells was performed by counting the number of positive cells for antigen staining per ten high power fields (HPFs). Each circle represents the murine lungs of one evaluated animal. Bars represent the mean and error bars represent the standard error of the mean. Number of virus antigen-positive cells in murine lungs were compared on 3, 10 and 21 dpi to the number of virus antigen-positive cells on 1 dpi by Kruskall-Wallis test with Dunn's multiple comparisons test. $*p < 0.05$, $***p < 0.001$. c) Comparison of the number of virus antigen-positive cells in murine lungs of xenografted and non-grafted NSG mice. Colored circles indicate the values for murine lungs of human lung xenografted animals, whereas grey circles indicate those of non-grafted animals. The dashed line indicates upper limit of detection (ULoD).
(TIF)

**S8 Fig. Human lung xenograft production of human cytokines and chemokines during orthohantavirus infection.** Heatmap displaying quantification of human cytokines and chemokines in human lung xenografts. The potentially relevant biological function of each cytokine and chemokine in context of orthohantavirus infection is indicated on the left. Tissue homogenates weighing less than 10 mg are excluded for analyses, while for inoculated animals only tissue homogenates positive for viral RNA were included. Cytokine levels were expressed as picogram cytokine/chemokine per gram tissue and inoculated samples harvested on 10 and 21 days post inoculation were compared to samples from mice that were left uninfected by Kruskall-Wallis test with Dunn's multiple comparisons test. Each cell represents the mean fold change of the cytokine/chemokine levels of tissues from inoculated animals over the respective mean levels from uninfected animals. Grey cells represent that all samples within a group were below lower limit of detection. $*p < 0.05$, $**p < 0.005$, $***p < 0.001$.
(TIF)

**S1 Table. Virus stocks and inoculations used during the experiment.**
(DOCX)

**S2 Table. Experimental set-up of the study.**
(DOCX)

**S3 Table. Experimental set-up of preclinical application of the model.**
(DOCX)

**S4 Table. Taqman primers and probes.**
(DOCX)

**S1 Data. Raw data.**
(XLSX)

## Acknowledgments

The authors thank Ingeborg van Middelkoop, Simone Stofberg, Ineke Maas and Vincent Duiverman for assistance with the animal studies, Melanie van Heteren-Hemelop for the primers and probes design, Ron Fouchier, Sander Herfst and Edwin Veldhuis Kroeze

(Department of Viroscience, Erasmus MC) for scientific discussions, and Sven Reiche (Department of Experimental Animal Facilities and Biorisk Management, Friedrich-Loeffler-Institut, Germany) for providing the antibody TULV 1.

## Author contributions

**Conceptualization:** Melanie Rissmann, Danny Noack, Barry Rockx.

**Data curation:** Melanie Rissmann, Danny Noack, Thomas M. Spliethof.

**Formal analysis:** Melanie Rissmann, Danny Noack.

**Funding acquisition:** Bart L. Haagmans, Marion P.G. Koopmans, Barry Rockx.

**Investigation:** Melanie Rissmann, Danny Noack, Thomas M. Spliethof.

**Methodology:** Melanie Rissmann, Danny Noack, Thomas M. Spliethof, Vincent P. Vaes, Rianne Stam, Peter van Run, Jordan J. Clark, Georges M.G.M. Verjans, Florian Krammer, Judith M.A. van den Brand.

**Project administration:** Melanie Rissmann, Barry Rockx.

**Resources:** Jordan J. Clark, Georges M.G.M. Verjans, Bart L. Haagmans, Florian Krammer, Barry Rockx.

**Software:** Danny Noack.

**Supervision:** Melanie Rissmann, Marion P.G. Koopmans, Judith M.A. van den Brand, Barry Rockx.

**Validation:** Melanie Rissmann, Danny Noack.

**Visualization:** Melanie Rissmann, Danny Noack.

**Writing – original draft:** Melanie Rissmann, Danny Noack, Barry Rockx.

**Writing – review & editing:** Melanie Rissmann, Danny Noack, Thomas M. Spliethof, Vincent P. Vaes, Rianne Stam, Peter van Run, Jordan J. Clark, Georges M.G.M. Verjans, Bart L. Haagmans, Florian Krammer, Marion P.G. Koopmans, Judith M.A. van den Brand.

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
