## [Decision Letter · Decision Letter 0]

7 Oct 2024

Dear Dr. Rissmann,

Thank you very much for submitting your manuscript "A pan-orthohantavirus human lung xenograft mouse model and its utility for preclinical studies" for consideration at PLOS Pathogens. As with all papers reviewed by the journal, your manuscript was reviewed by members of the editorial board and by several independent reviewers. In light of the reviews (below this email), we would like to invite the resubmission of a significantly-revised version that takes into account the reviewers' comments.

While all three reviewers acknowledge that the human xenograft lung model has merits for hantavirus research, at least two of the reviewers indicated that the utility of model is being "oversold". Please heed these comments in toning down the conclusions and also provide a more balanced review of the pros and cons of your current model using immunocompromised mice with extant immunocompetent mouse models or human organic systems.        

We cannot make any decision about publication until we have seen the revised manuscript and your response to the reviewers' comments. Your revised manuscript is also likely to be sent to reviewers for further evaluation.

Sincerely,

Benhur Lee

Section Editor

PLOS Pathogens

Benhur Lee

Section Editor

PLOS Pathogens

Michael Malim

Editor-in-Chief

PLOS Pathogens

orcid.org/0000-0002-7699-2064

Reviewer's Responses to Questions

**Part I - Summary**

Reviewer #1: Rissman et al present a novel in vivo-model for studies of hantaviruses and for therapeutics against them. The presented model is interesting and can hopefully be of use for the future establishment of treatments, which is highly needed.

Strengths: There is a lack of small animal models for hantaviruses, hampering studies of treatment and basic hantavirus research. This model can provide a first step towards such a model by including human lung tissue.

Weaknessess: The model is based on immunosuppressed mice, hence important factors contributing to the defence against hantavirus are lacking as are potential immunopathogenic effects. This likely has only a limited impact on studies on for example the effect of anti-hantavirus monoclonal antibodies and on viral replication in human lung cells, but it can limit the possible use of this model for other purposes.

Novelty: The xenograft model has previously been used for other viruses, a novelty is that it is here adopted to several different hantaviruses allowing for studies comparing different hantaviruses.

General execution and scholarship. The experiments are well designed. Results/data are presented in a clear way in text and figures.

Reviewer #2: The report by Rissmann et al. describes characterization of a lung xenograft mouse model for orthohantaviruses. They show that the subcutaneously grafted tissue can support viral replication, and that virus can spread to other uninoculated graphs as well as murine tissues. The authors also present cytokine analyses which provide some depth to the investigations and appear to correlate with previous literature both in humans and animal models. Importantly, the kinetics of replication (increasing over time until study end) and absence of histopathological findings differentiate these mice from a disease model; the xenograft mouse model presented here is an infection model with the advantage of incorporating human tissue evaluation in a model system. The inclusion of appropriate controls (non-grafted mice) indicating that grafting does not alter overall susceptibility to infection but does increase titers and results in more efficient viral spread further supports use primary as an infection model. However, infection models can serve an important role, particularly when replication can achieve levels that permit more robust evaluation of countermeasures, which is supported by work presented in the report. Overall, the studies are well performed and provide interesting data to the field. It was nice to see both HCPS- and HFRS-associated viruses included. I would recommend the authors revisit the text to add more substantial details when discussing the literature both in the intro and discussion (info is often very high level/vague vs. citing specific details of the literature). In addition, the report would greatly benefit from text highlighting the limitations of the system (and caveats of data interpretation). I don’t think this would detract from the advancement presented, but would greatly improve the utility of the report, and aid readers in consideration for how these data could be applied and for future studies with these mice.

Reviewer #3: This manuscript addresses a long-standing issue in the hantavirus field by generating a novel human lung xenograft mouse model that supports infection and replication of HCPS and HFRS-causing viruses. Subcutaneous implantation of human fetal lung tissues in subcutaneous pockets of NSG mice led to successful engraftment and vascularization of xenografts that represented key cell types of adult human lungs, including endothelial cells, epithelial cells, and macrophages, which are targets of hantavirus infection. These xenografts supported the replication of diverse hantaviruses upon intragraft inoculation. Hantavirus infections caused viremia; viruses were disseminated to non-inoculated human lung xenografts and induced localized chemokine and cytokine responses, including IP-10 and RANTES expression. This model can be used for the preclinical evaluation of anti-hantavirus therapeutics as demonstrated by the blockade of Andes virus infection and dissemination by a neutralizing antibody. Although primarily descriptive and hantavirus disease is not recapitulated in this model, this is a well-planned and executed study describing a significant advancement in the field. The manuscript is well-written, and the conclusions are well-supported by the data.

**Part II – Major Issues: Key Experiments Required for Acceptance**

Reviewer #1: 1. The authors provide a good discussion regarding limitations of lung organoids (lines 330-346). However, I lack a discussion about possible limitations of the xenograft mice model compared to organoids, and of possible similarities/differences between these two different systems.

This is important as this novel model seems appropriate for studies of certain research questions, such as early replication in a human lung model and test of neutralizing antibodies. However, the xenografts do not function as lungs as there is no air-organ interface, hence the early steps of infection (aerosol transmission and initial infection of target cells in the lung) will not be the same as in human patients. This should be briefly discussed as a limitation of the study. Moreover, as the model is based on immunocompromised mice it needs to be discussed how this model relates to other animal models. For example, most likely lab mice can be asymptomatically infected with hantaviruses, which seems to be verified in this manuscript (Supplementary Fig 5/Lines 180-182) – What is the benefit of this xenograft model based on immunosuppressed mice compared to earlier mice models based on immunocompetent lab mice-strains?

2. Fig 1D/Line 123 and Fig 5/Line223. Are the resident immune cells/macrophages of human or mice original?

3. The finding of hantavirus-specific cytokine responses in the human lung (Lines 231-245) is intriguing, it would be interesting to see levels of human and mouse cytokines in mouse serum/plasma. This could provide more information regarding if the hantavirus-mediated effects on cytokines is limited to the xenografts or if there is also a systemic effect via infected/stimulated mouse cell-produced cytokines. An additional control could be infection of human endothelial cells in vitro with the different hantaviruses for comparison of the results obtained in the xenografts.

Reviewer #2: No major issues.

Reviewer #3: None.

**Part III – Minor Issues: Editorial and Data Presentation Modifications**

Reviewer #1: 1. It should be more clearly stated in the beginning of the manuscript that the model is based on immunosuppressed mice.

2. In figure legends, include data on how many mice was included in each experiment (as several xenografts can come from the same mouse).

Reviewer #2: Line 94: provide more details on models that result in disease and those that demonstrate replication in absence of clinical signs/pathology either here or in discussion, as this is relevant to the data presented in the xenograft model. Also, Andes virus was reported to cause disease in immunocompetent hamsters (PMID: 11601912) – this is one of the references listed. If the authors are trying to emphasize models that can be used for both HCPS- “and” HFRS, this can be written more clearly with some more context by providing details of the reports to date.

Line 115: Specify location of subcutaneous implants (dorsum?). For this comment and some that follow, the paper would benefit from adding or moving some of the details from the material and methods to the main text.

Line 144: For clarity, please add more details in the text regarding the study design, including groups sizes, serial timepoints and end of study timepoint.

Line 196: “to variable extents” please provide some more details in the text. Within tissue types, between tissues, between individuals in the experimental group…

Line 247 – 280: based on viral kinetics presented earlier in the paper, the highest levels were typically observed at d21 (the latest timepoint) – is d10 sufficient or would these models require evaluation at later timepoints also? Are there concerns about a “persistent” infection phenotype being observed in the lung xenografts vs. modeling acute infection in “physiologically relevant” tissues?

Line 262: “no viral RNA was found in the sera of KL-AN-5E8 treated animals” – please specify at what timepoints this was assessed

Line 275: Lung xenografts were evaluated in treated mice, were murine tissues also evaluated?

Line 365, 367: should be “Tecniplast” without the “h”?

Line 369: “Sterilized” food and water?

Line 373: clarify if mice were euthanized by exsanguination and/or isoflurance overdose (followed by cervical dislocation?)

Line 375: Were inocula backtitered? Please provide target and backtiter doses

Discussion: Please add a section of limitations of the model, caveats to data interpretation, thoughts on replication kinetics (what timepoints

Fig 3: Please clarify group sizes in the figure legend as only mean is depicted (group sizes would be helpful in other legends also, but more critical for this one).

Other:

-Sundström et al. reported on virus kinetics and associated cytokine response in a 3D Human Lung Tissue Model (PMID: 26907493; PMCID: PMC4764364). This seems like a potentially overlooked relevant reference to include in this paper?

-There are many references on cytokine analyses in cells, animal models and humans – it would strengthen the paper to discuss if how these results are similar or differ to previous studies and the resultant clinical relevance.

Reviewer #3: Include a sentence about the success rate of xenografting.

Fig 1b - Add scale bar.

Fig S2 legends - “azoom-in” should be “a zoom-in”

PLOS authors have the option to publish the peer review history of their article (what does this mean? ). If published, this will include your full peer review and any attached files.

**Do you want your identity to be public for this peer review?** For information about this choice, including consent withdrawal, please see our Privacy Policy .

Reviewer #1: No

Reviewer #2: No

Reviewer #3: No

Figure Files:

Data Requirements:

Please note that, as a condition of publication, PLOS' data policy requires that you make available all data used to draw the conclusions outlined in your manuscript. Data must be deposited in an appropriate repository, included within the body of the manuscript, or uploaded as supporting information. This includes all numerical values that were used to generate graphs, histograms etc.. For an example see here on PLOS Biology: http://www.plosbiology.org/article/info%3Adoi%2F10.1371%2Fjournal.pbio.1001908#s5 .
---

## [Editor Report · Decision Letter 1]

1 Jan 2025

Dear Dr. Rissmann,

We are pleased to inform you that your manuscript 'A pan-orthohantavirus human lung xenograft mouse model and its utility for preclinical studies' has been provisionally accepted for publication in PLOS Pathogens.

Best regards,

Benhur Lee

Section Editor

PLOS Pathogens

Benhur Lee

Section Editor

PLOS Pathogens

Sumita Bhaduri-McIntosh

Editor-in-Chief

PLOS Pathogens

orcid.org/0000-0003-2946-9497

Michael Malim

Editor-in-Chief

PLOS Pathogens

orcid.org/0000-0002-7699-2064
---

## [Editor Report · Acceptance letter]

Dear Dr. Rissmann,

We are delighted to inform you that your manuscript, "A pan-orthohantavirus human lung xenograft mouse model and its utility for preclinical studies," has been formally accepted for publication in PLOS Pathogens.

Best regards,

Sumita Bhaduri-McIntosh

Editor-in-Chief

PLOS Pathogens

orcid.org/0000-0003-2946-9497

Michael Malim

Editor-in-Chief

PLOS Pathogens

orcid.org/0000-0002-7699-2064